# Outcome Measures and Biomarkers for Clinical Trials in Hereditary Spastic Paraplegia: A Scoping Review

**DOI:** 10.3390/genes14091756

**Published:** 2023-09-03

**Authors:** Sue-Faye Siow, Dennis Yeow, Laura I. Rudaks, Fangzhi Jia, Gautam Wali, Carolyn M. Sue, Kishore R. Kumar

**Affiliations:** 1Sydney Medical School, University of Sydney, Camperdown 2050, Australiac.sue@neura.edu.au (C.M.S.);; 2Department of Clinical Genetics, Royal North Shore Hospital, St Leonards 2065, Australia; 3Neuroscience Research Australia, University of New South Wales, Randwick 2031, Australia; 4Rare Disease Program, Garvan Institute of Medical Research, Darlinghurst 2010, Australia; 5Translational Neurogenomics Group, Molecular Medicine Laboratory and Department of Neurology, Concord Hospital, Concord 2139, Australia; 6Neurodegenerative Service, Prince of Wales Hospital, Randwick 2031, Australia; 7School of Clinical Medicine, UNSW Medicine & Health, University of New South Wales, Kensington 2052, Australia

**Keywords:** hereditary spastic paraplegia, outcome measures, biomarkers, clinical trials, scoping review

## Abstract

Hereditary spastic paraplegia (HSP) is characterized by progressive lower limb spasticity. There is no disease-modifying treatment currently available. Therefore, standardized, validated outcome measures to facilitate clinical trials are urgently needed. We performed a scoping review of outcome measures and biomarkers for HSP to provide recommendations for future studies and identify areas for further research. We searched Embase, Medline, Scopus, Web of Science, and the Central Cochrane database. Seventy studies met the inclusion criteria, and eighty-three outcome measures were identified. The Spastic Paraplegia Rating Scale (SPRS) was the most widely used (27 studies), followed by the modified Ashworth Scale (18 studies) and magnetic resonance imaging (17 studies). Patient-reported outcome measures (PROMs) were infrequently used to assess treatment outcomes (28% of interventional studies). Diffusion tensor imaging, gait analysis and neurofilament light chain levels were the most promising biomarkers in terms of being able to differentiate patients from controls and correlate with clinical disease severity. Overall, we found variability and inconsistencies in use of outcome measures with a paucity of longitudinal data. We highlight the need for (1) a standardized set of core outcome measures, (2) validation of existing biomarkers, and (3) inclusion of PROMs in HSP clinical trials.

## 1. Introduction

Hereditary Spastic Paraplegia (HSP) refers to a group of inherited neurodegenerative conditions characterized by lower limb spasticity and weakness. HSP is rare, with a prevalence of 0.3 to 5.5 per 100,000 people, depending on country [1,2,3,4]. HSP is associated with significant disability and a negative impact on quality of life [5]. There are over 80 recognized HSP-associated genes with broad phenotypic variability [6,7,8]. Clinically, HSP can be categorized into pure HSP—symptoms limited to weakness, spasticity, impaired vibration sense in the lower limbs, and bladder dysfunction; and complex HSP—where there are additional neurological and non-neurological manifestations [9]. The phenotypic and genotypic heterogeneity of HSP and the rarity of the condition pose a challenge to the development of suitable outcome measures and biomarkers.

Advances in genetic testing have led to the rapid discovery of genes associated with HSP [10], accelerating the discovery of therapeutic targets. Patient-derived stem cell and animal models have identified potential drug treatment candidates targeting underlying pathogenesis for specific HSP genotypes, such as noscapine for HSP-*SPAST* [11,12,13,14,15]. Currently, the rate of drug discovery is far outpaced by the rate of gene discovery for HSP and a shift in HSP research to developing treatment options is required [10,16,17]. Outcome measures and biomarkers that can measure the efficacy of therapeutic interventions in clinical trials are needed to facilitate this.

The choice of appropriate outcome measures in interventional trials is critical to demonstrating a meaningful treatment effect [18]. Initiatives such as COnsensus-based Standards for the selection of health Measurement INstruments (COSMIN) and Core Outcome Measures in Effectiveness Trials (COMET) aim to guide the selection of appropriate outcome measures [19]. There is currently a lack of standardized outcome measures used in HSP clinical trials, with recent reviews of interventional trials in HSP showing heterogeneity of outcome measures [17,20]. Inconsistency of outcome measures leads to an increased risk of reporting bias due to post hoc selection of outcomes based on results rather than the use of pre-specified primary outcomes [21]. Furthermore, use of different outcome measures limits comparison and meta-analysis of results from different studies [21].

There are no current consensus guidelines for outcome measures in HSP clinical trials. To address this, we performed a scoping review of outcome measures and biomarkers in HSP to identify suitable outcome measures, provide recommendations for future HSP clinical trials, and identify areas for further research.

## 2. Materials and Methods

### 2.1. Search Strategy

This study was conducted according to published guidelines for conducting a systematic scoping review [22,23]. Under the guidance of an academic librarian, we performed a search in Embase, Medline, Scopus, Web of Science and the Central Cochrane databases using the search terms “Hereditary Spastic Paraplegia”, “biomarker”, “outcome measure”, and “patient reported outcome measure” (Appendix B). Additional studies were identified by searching the references of the included articles and relevant review articles.

### 2.2. Selection Criteria

We kept the selection criteria broad to capture all possible outcome measures and biomarkers. The inclusion criteria included studies involving patients with HSP of any age and gender, and that included a description of the outcome measures or biomarkers for HSP. Abstracts were included if novel outcome measures or biomarkers were described.

We excluded review articles, single-case reports, trial protocols with no published results, abstracts with no novel outcome measures/biomarkers, and articles that did not involve humans or human samples, did not include outcome measures/biomarkers, or were not in English.

### 2.3. Screening of Search Results

The screening and data extraction process was conducted using Covidence, a web-based collaboration software platform that streamlines the production of systematic and other literature reviews [24]. The first author (S.F.S.) excluded all irrelevant results—studies unrelated to HSP or not in English and duplicate studies. Authors S.F.S. and D.Y. independently reviewed the abstracts according to the selection criteria for the first 50 results. Authors S.F.S. and D.Y. independently reviewed the remaining abstracts, applying the finalized exclusion criteria.

### 2.4. Data Extraction and Analysis

Authors S.F.S., D.Y., L.R., and F.J. reviewed two full texts each to test the data extraction template. The template was modified by discussion at a team meeting. The team then reviewed six full texts, each with each author reviewing the same three articles as two other authors (~30% overlap) and met to resolve any discrepancies and standardize the extraction process. The rest of the studies were reviewed by one reviewer (D.Y., L.R., or F.J.) and verified by a second reviewer (S.F.S.). At each stage, discrepancies were resolved through a team meeting. The data were extracted using a standardized data extraction template, including information on study characteristics, aim of study, participant characteristics, characteristics of interventions, outcome measures, and ability of outcome measure to (i) distinguish patients versus controls, (ii) demonstrate change over time; (iii) show response to the intervention; and (iv) correlate with other measures. Study quality analysis was not performed, as is usual for scoping reviews [22].

Data were analyzed descriptively to provide an overview of study characteristics and outcome measures. Outcome measures were grouped according to the types of clinical outcome assessments as defined by the U.S. Food and Drug Administration [25]:Clinician-reported outcome measures (CROM): measurement of clinical signs or findings performed by a health professional.Performance outcome measures (PerfOM): measurement with a standardized task, either administered by a trained individual or undertaken by the patient without assistance.Patient-reported outcome measures (PROM): measurement of patient-reported health status.


Biomarkers were grouped according to the assessed modality:
Laboratory-based biomarkers;Neuroimaging biomarkers;Neurophysiology biomarkers;Other biomarkers.

As other groups have previously published reviews of non-randomized interventional clinical trials [17,20], we chose to perform further analysis of randomized controlled trials (RCTs) to compare the outcome measures used, as RCTs are the study type with the highest level of evidence for treatment effectiveness [26].

## 3. Results

### 3.1. Search Results

A total of 1930 search results were identified, and 1437 abstracts were screened after duplicates were removed, of which 1222 were identified as irrelevant. The full texts of the remaining results (*n* = 215) were assessed according to inclusion criteria. Ten more studies were identified by searching the references of the included articles (Figure 1).

### 3.2. Study Characteristics

A total of 70 studies that met the inclusion criteria were identified. These studies were published between 1991 and 2022; date limits were not set to include as many studies as possible.

The majority (78.6%) of the included studies were observational studies, and only a quarter (25.7%) of studies were interventional studies. Most studies did not include longitudinal data (75.7%), and over half did not include a control group (51.4%). Participant genotype was predominantly mixed or unknown (65.7%), and sample sizes were small (mean of 36.99 participants) (Table 1). A list of all studies with relevant details is included as Appendix A.

We assessed each outcome measure for (1) the ability to differentiate patients versus controls if a control group was included, (2) the ability to demonstrate disease progression if longitudinal data were included, (3) the ability to demonstrate response to the intervention if an intervention was assessed, and (4) any correlation with other biomarkers or outcome measures. Although we report the number of studies that fulfilled the criteria for (1), (2) and (3) to illustrate the available evidence for each outcome measure, it is important to note that (1) in studies with multiple outcome measures, the control group was compared to the patient group for some but not all outcome measures, (2) not all longitudinal studies were designed to assess the ability of an outcome measure to measure disease progression, (3) an outcome measure may not demonstrate a response to the intervention for many reasons including efficacy of intervention, duration of trial, and timing of the outcome measure relative to the intervention.

### 3.3. Clinical Outcome Assessments

#### 3.3.1. Clinician Reported Outcome Measures

The Spastic Paraplegia Rating Scale (SPRS) was the most widely used CROM, reported in twenty-seven studies (Table 2). Although 12/27 included studies control groups, a comparison of SPRS scores in patients versus controls was performed in only two studies, both showing significant differences [28,29]. It is important to note that SPRS values from healthy controls are most relevant when comparing to pre-symptomatic HSP carriers rather than for comparison to individuals with symptomatic HSP. 8/27 studies were longitudinal, and only six of those studies showed disease progression over time with the SPRS (median follow-up time 12–31 months) [30,31,32,33,34,35]. The SPRS was used as an outcome measure in 4/18 interventional studies and showed a response to the intervention in only one of the four studies [36]. The SPRS was commonly used to correlate with other biomarkers or outcome measures (21/27 studies) (see Appendix A).

The SPATAX-EUROSPA disability score, another HSP-specific CROM, was less commonly used than the SPRS (Table 2). There were no control data and change over time was studied in only one study showing disease progression in 3/31 patients included in the study [32]. It was used in one interventional study [37] and did not show any significant change with intervention.

CROMs developed for other neurological conditions and generic functional CROMs were also used to assess patients with HSP. The Scale for Assessment and Rating of Ataxia (SARA) was used in three studies and showed a difference between patient versus controls in 1/3 studies [38] and longitudinal change in 1/3 studies [39]. The Amyotrophic Lateral Sclerosis rating scale revised (ALSFRS-R) was used in two studies [40,41], showing disease progression in one but not the other, and was shown to correlate with serum and CSF neurofilament heavy chain [40,41], though one paper included patients with ALS in their analysis (Appendix A).

#### 3.3.2. Performance Outcome Measures

PerfOMs assessing motor function were widely used in the included studies, with twenty-one different outcome measures identified. The most commonly reported motor PerfOMs were the modified Ashworth Scale (MAS) (18 studies) and the 10 m walk test (10MWT), 6 min walk test (6MWT) and their variations (14 studies) (Table 3). Compared to CROMs, motor PerfOMs were more commonly used as outcome measures for interventional studies—MAS in 13/18 interventional studies, 10MWT/6MWT/variations in 9/18 interventional studies. Conversely, motor PerfOMs were less commonly used to correlate with other outcome measures or biomarkers, 2/18 studies for MAS and 3/14 studies for 10MWT and its variations (Appendix A). Similar to the CROMs, there were very little longitudinal data—3/18 for MAS, 1/14 for 10MWT and variations.

PerfOMs measuring cognition were used in ten studies for descriptive purposes only rather than to measure outcomes in interventional studies.

#### 3.3.3. Patient Reported Outcome Measures

Most PROMs used assessed quality of life (12/17), while others assessed fatigue, pain, and autonomic symptoms (Table 4). The most used PROM was the Short Form Health Survey-36 (SF-36) and its derivative, SF-12, in seven studies, followed by EuroQoL-5 Dimensions (EQ-5D) in three studies. PROMs were not commonly used in interventional studies (5/18 interventional studies). There was no longitudinal data for PROMs in the included studies. The SF-36 and its derivatives, EQ-5D, Becks Depression Inventory (BDI-V), Zung depression score, Brief Pain inventory, Modified Fatigue Impact Scale and multidimensional fatigue inventory showed differences between patients and controls in some but not all studies.

### 3.4. Biomarkers

#### 3.4.1. Laboratory-Based Biomarkers

Serum and cerebrospinal fluid (CSF) neurofilament light chain (NfL) levels, and serum and CSF 25- and 27-hydroxycholesterol (25- and 27-OHC) levels were the two most studied biochemical biomarkers—included in six and four studies, respectively (Table 5). Serum and CSF NfL levels were able to differentiate patient vs. control groups or historical control values in all studies. Serum and CSF NfL levels were shown to correlate with SPRS scores in two studies [29,42] but this correlation was not present in the other two studies [32,43] (Appendix A). Plasma and CSF 25- and 27-OHC were significantly elevated in individuals with HSP-*CYP7B1* (SPG5) compared to controls or reported normative values in all four studies [35,43,44,45]. In the few longitudinal studies, there was no change over time in NfL levels [32] or 25- and 27-OHC levels [35,44]. NfL levels were not used as outcome measures in any interventional studies; however, 25- and 27-OHC levels showed a response to treatment with statins in interventional studies [35,44]. Other biochemical markers reported, such as lipidomics, amino acid levels, mitochondrial DNA levels and cell morphomics, were more commonly used as diagnostic biomarkers rather than to measure disease progression or response to the intervention.

#### 3.4.2. Neuroimaging Biomarkers

Magnetic resonance imaging (MRI) of the brain and spine, with or without volumetric analysis, was used in seventeen studies, diffusion tensor imaging (DTI) in seven studies and magnetic resonance spectroscopy (MRS) in four studies (Table 6). MRI brain and spine differentiated patients vs. controls in six studies and showed longitudinal change in only one study [46]. No imaging parameters were used as outcome measures for interventional studies. MRI findings were found to correlate with SPRS scores in 3/4 of the studies, and DTI findings correlated with SPRS scores in 2/3 of the studies (Appendix A).

#### 3.4.3. Neurophysiology Biomarkers

Motor evoked potentials (MEPs) and nerve conduction studies/electromyography (NCS/EMG) were the most widely used neurophysiological markers, being used in ten and nine studies, respectively (Table 7). Although MEPs were used as outcome measures in three interventional studies, no changes in MEP results were demonstrated in response to any of the interventions. NCS/EMG, somatosensory evoked potentials (SSEPs), visual evoked potentials (VEPs) and brainstem auditory evoked potentials (BAEPs) were mostly used as descriptive measures with few studies using these measures to compare patients versus controls, study longitudinal cohorts, or to evaluate interventions.

#### 3.4.4. Other Biomarkers

We identified eight studies that utilized gait analysis—seven were laboratory-based, and one was a mobile system [34] (Table 8). Four studies described the use of an infrared multi-camera motion analysis system [28,47,48,49], one used pressure sensors [50], and another two did not provide details of the gait analysis system used [51,52] (Appendix A). Four interventional studies used gait analysis as an outcome measure, with only two of those studies showing a response to the intervention [49,50]. Gait analysis was able to differentiate patients with HSP from patients with spastic diplegia [47], healthy controls [34,48] and pre-symptomatic HSP-*SPAST* carriers [28]. Gait parameters also correlate with SPRS scores [28,34] (Appendix A).

SD-OCT was investigated in four studies, with one study showing statistically significant retinal nerve fiber layer (RNFL) thinning in patients compared to normative values but no change over time (Table 8).

Video-oculography and rotational chair testing and speech assessment showed differences between patients vs. controls [53,54].

#### 3.4.5. Genotype-Specific Biomarkers in HSP

We identified biomarkers that were specific to particular HSP genotypes (Table 9). The most widely studied were serum and CSF oxysterols, 25- and 27-hydroxycholesterol (25- and 27-OHC) levels, in HSP-*CYP7B1* or SPG5. 25- and 27-OHC levels were significantly elevated in the serum and CSF of patients compared to healthy controls. Plasma and serum oxysterol levels were used as outcome measures in two clinical trials showing reduced levels with atorvastatin treatment but no clinical correlation [35,44]. Most genotype-specific biomarkers were reported in single studies, with some studies (citrulline, lipidomics, glycosylceramide profile, and scanning electron microscopy of hair shafts) having small patient numbers (*n* ≤ 5).

### 3.5. Randomized Controlled Trials in HSP

We identified eight randomized controlled trials published over two decades (Appendix A). Of note, 5/8 RCTs we reviewed were not included in previously published reviews.

Six studies used a crossover design, while two studies were parallel randomized trials. Only two studies recruited patients with a single specific HSP genotype; the other six included mixed and unknown genotypes. All studies had small sample sizes (range 8–49). Two studies did not define a primary outcome measure [37,62]. Four studies showed a positive response to treatment. The outcome measures were heterogeneous, with no consistency between studies despite six studies aiming to evaluate spasticity in response to the intervention. The most common outcome measures used were the 10MWT and modified Ashworth score, with two positive studies showing a significant improvement in MAS with treatment [62,63]. Study durations were generally short—only one study had a duration of more than 6 months [64]. There was no clear association between the duration of the trial and response to the intervention.

## 4. Discussion

In this scoping review, we compiled a comprehensive list of outcome measures used in HSP (Figure 2) and categorized them according to construct (Table 2, Table 3, Table 4, Table 5, Table 6, Table 7 and Table 8), with information on key measurement properties. This is an important resource to inform the choice of outcome measures for future clinical trials in HSP. We identified eighty-three outcome measures highlighting the heterogeneity and inconsistency of outcome measures used. We identify a need for standardized outcome measures and recommendations for use in clinical trials, such as core outcome sets (COS) developed for other neurological conditions [19,65]. We identify common limitations of the included studies in this review (Figure 3) and list the advantages and disadvantages of commonly used outcome measures (Table 10).

### 4.1. Recommendations for Future Research

#### 4.1.1. Choice of Outcome Measure

This scoping review is the first comprehensive reference of outcome measures available for HSP according to outcome/construct. Based on our findings, we propose that a COS include a CROM—SPRS, PROM—SF36, and an objective biomarker—DTI, gait analysis or serum/CSF NfL.

We recommend the SPRS as it was the most commonly used CROM in HSP-related studies, has undergone psychometric testing in a cohort of individuals with HSP [67], has demonstrated cross-cultural validity [68], has been tested for responsiveness in longitudinal and interventional studies [30,33,35,62,64,69,89], and is a disease-specific outcome measure. In addition, it includes an assessment of motor function, and therefore, a performance outcome measure (PerfOM) for motor function is not required. However, we note that SPRS was not designed for use in the pediatric population, and therefore, we identify a need for a validated pediatric HSP-specific CROM.

We recommend the SF-36 as the most suitable PROM as it was commonly used in HSP-related studies, is well-validated in healthy controls and other conditions [72], has cross-cultural validity in the HSP population [73,74], and correlates with disease severity as measured with the SPRS [75,90,91] and gait analysis [34]. However, we note that the SF-36 is a generic QoL measure and may not be sensitive to smaller changes in HSP-specific symptoms, such as spasticity and bladder function, that are likely key targets for intervention in future HSP trials [92]. Therefore, we identify a need for an HSP-specific QoL scale to address the need for more sensitive tools, particularly when evaluating small changes in response to treatment in a slowly progressive condition. These findings are echoed by a recent study of CROMs and PROMs in HSP, identifying the SPRS as a suitable CROM to measure disease progression and the need for an HSP-specific PROM [89].

Diffusion tensor imaging, gait analysis, and neurofilament light chain levels are promising objective biomarkers likely to be suitable for use in clinical trials. However, further studies are needed to establish the sensitivity and specificity of these biomarkers in HSP, including longitudinal studies. Biomarkers identified through knowledge of disease pathways in specific HSP genotypes can aid in the diagnosis and measurement of treatment response. Abnormal plasma and CSF 25- and 27-OH levels are seen in individuals with HSP-*CYPB1* (SPG5) and responded to treatment with statins [35,44]. More recently, two groups used different approaches to differentiate individuals with HSP-*SPAST* (SPG4) from healthy controls by analyzing peripheral blood mononuclear cells. Our group demonstrated reduced levels of acetylated α-tubulin seen on flow cytometry [93], while another group showed increased distance between cell and nucleus centroids on automated image analysis [94]. Both studies used surrogate markers of microtubule dysfunction based on the known role of spastin in regulating microtubule dynamics in HSP-*SPAST*. These findings highlight the need for further research to identify genotype-specific biomarkers that are more likely to be sensitive and specific for particular HSP genotypes. For the evaluation of existing outcome measures or the development of new outcome measures, we recommend referring to published guidelines [19] to ensure validity and reliability. Development of a COS for HSP should be informed by published recommendations, COS-STAD, to ensure that the included outcome measures meet minimum standards [95].

#### 4.1.2. Recommendations for Trial Design

Recommendations for HSP clinical trial design will improve the quality and consistency of reporting of evidence to strengthen conclusions drawn from clinical trials. We identify a need for larger sample sizes in HSP studies, a particular challenge when studying rare conditions. International, multi-center collaborations are a potential solution to sample size challenges and have the added benefit of cross-cultural applicability of study results. It is important for outcome measures chosen to be available in different languages, relevant in different populations, and feasible for use in low-resource settings. Biomarkers that can be collected at recruitment sites and analyzed in a central facility are ideal for multi-center trials to improve the standardization of data. Although studies of single HSP genotypes are ideal for consistency within target populations, this may not be feasible when trying to attain large sample sizes. Therefore, a grouping of HSP genotypes or other neurodegenerative disorders according to similar underlying disease pathophysiology may allow for more efficient research of targeted therapeutic agents. Clinical trials for HSP may require longer periods of treatment and follow-up to demonstrate significant treatment effects, particularly in the more slowly progressive genotypes, such as HSP-*SPAST*. Using a combination of multiple outcome measures and biomarkers can account for phenotypic variability, even within the same genotype. It may improve the chance of detecting a treatment effect signal in heterogeneous patient cohorts.

#### 4.1.3. Study Limitations

A limitation of this study was the use of the search terms “outcome measure” and “biomarker”, which inevitably missed relevant studies [19]. When planning our study design, we considered the volume of search results from individual searches for each outcome, e.g., spasticity, mobility, neuroimaging, etc. and deemed that approach unfeasible.

Systematic reviews on specific outcome measures have been published previously [81,86,96,97] and provide information on the utility of these outcome measures. Due to the nature of a scoping review, we were unable to perform qualitative analysis of each outcome measure to assess validity and reliability. Therefore, a systematic review specific to an outcome measure is required to answer this question. Our review identifies outcome measures that require further validation and provides an overview of the currently available literature surrounding the identified outcome measures.

## 5. Conclusions

In this scoping review, we present a critical assessment of currently available outcome measures for use in HSP clinical trials. We discuss the benefits and limitations of commonly used outcome measures and propose areas for further research. Given the emergence of multiple candidate HSP therapies in recent years [35,44,98], there is an urgent need for further development of a core set of validated and standardized outcome measures for use in HSP clinical trials to test the efficacy of these therapies.

## Figures and Tables

**Figure 1 genes-14-01756-f001:**
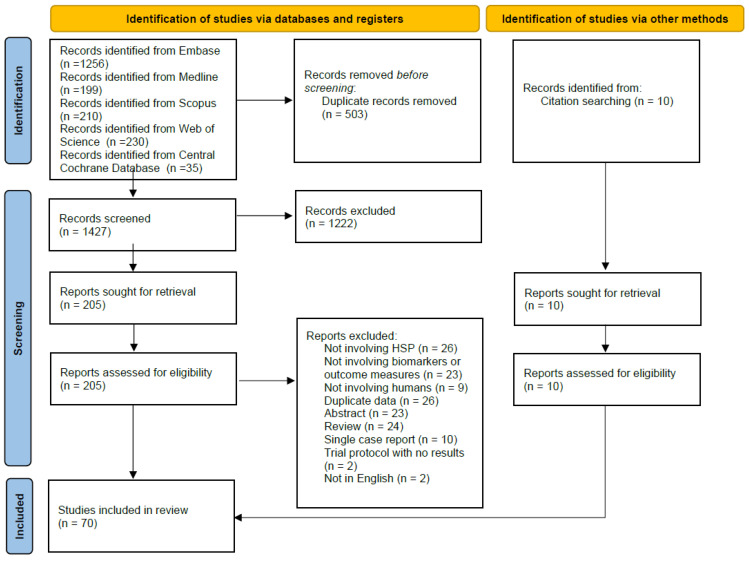
PRISMA flow chart [27].

**Figure 2 genes-14-01756-f002:**
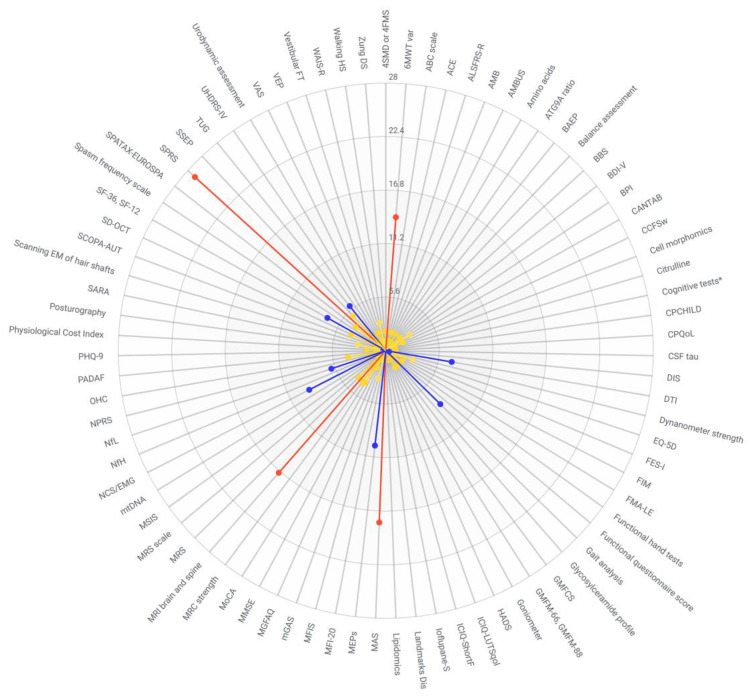
Outcome measures for hereditary spastic paraplegia according to frequency reported (Yellow 1–5 studies, Blue 6–10 studies, Red >10 studies). 4SMD or 4FMS—Four-stage functional scale of motor disability; 6MWT var—Six minute walk test and variations; ABC scale—Activities-specific Balance Confidence Scale; ACE—Addenbrooke’s Cognitive Exam; ALSFRS-R—Amyotrophic Lateral Sclerosis Rating Scale Revised; AMB—Ambulatory score; AMBUS—Distance walked in meters walked in 5 s without help; ATG9A ratio—Autophagy-related protein 9A ratio; BAEP—brainstem auditory evoked potentials; BBS—Berg Balance Scale; BDI-V—Becks Depression Inventory; BPI—Brief Pain Inventory; CANTAB—CANTAB cognitive assessment; CCFSw—Composite Cerebellar Functional Severity Score; Cognitive tests*—Verbal Learning and Memory Test, Farbe-Wort-Interferenz Test, Trail Making Test Part A and B, Frontal Assessment Battery, revised d2 Test of attention, Regensburg Word Fluency Test; CPCHILD—Caregiver Priorities and Child Health Index of Life with Disabilities; CPQoL—Cerebral Palsy quality of life questionnaire; DIS—Disability score; DTI—diffusion tensor imaging; EQ-5D—EuroQoL 5 Dimensions; FES-I—Falls Efficacy Scale-International; FIM—Functional Independence Measure; FMA-LE—Lower extremity subclass of Fugl–Meyer assessment; GMFCS—Gross Motor Function Classification Score; GMFM—Gross Motor Function Measure; HADS—Hospital Anxiety and Depression Scale; ICIQ-LUTSqol—International Consultation of Incontinence Questionnaire lower urinary tract symptoms quality of life; ICIQ-ShortF—International Consultation of Incontinence Questionnaire Short Form; Ioflupane-S—Ioflupane-single photon emission computed tomography (SPECT); Landmarks Dis—Landmarks of Disability; MAS—Modified Ashworth Scale; MEPs—motor evoked potentials; MFI-20—Multidimensional Fatigue Inventory; MFIS—Modified Fatigue Impact Scale; mGAS—Modified Goal Attainment Scale; MGFAQ—Modified version of Gillette Functional Assessment Questionnaire; MMSE—Mini-Mental State Exam; MoCA—Montreal Cognitive Assessment; MRC—Medical Research Council muscle strength; MRI—magnetic resonance imaging; MRS—magnetic resonance spectroscopy; MRScale—Modified Rankin Scale; MSIS—Multiple Sclerosis Impairment Scale; mtDNA—mitochondrial DNA load; NCS/EMG—nerve conduction studies/electromyography; NfH—Neurofilament heavy chain; NfL—Neurofilament light chain; NPRS—Numeric rating scale for pain; OHC—25 and 27 hydroxycholesterol; PADAF—protocol for evaluation of acquired speech disorder; PHQ-9—Patient Health Questionnaire; SARA—Scale for Assessment and Rating of Ataxia; scanning EM of hair shafts—scanning electron microscopy of hair shafts; SCOPA-AUT—Scale for Outcomes in Parkinson’s Disease for Autonomic Symptoms; SD-OCT—spectral domain optical coherence tomography; SF-36, SF-12—Short Form 36, Short Form 12; SPRS—Spastic Paraplegia Rating Scale; SSEP—Somatosensory evoked potentials; TUG—Timed Up-and-Go test; UHDRS-IV—Unified Huntington’s Disease Rating Scale Part IV; VAS—Visual Analogue Score; VEP—visual evoked potentials; Vestibular FT—video-oculography and rotational chair testing; WAIS-R—Wechsler Adult Intelligence Scale-revisited; Walking HS—Walking Handicap Scale; Zung DS—Zung Depression Scale.

**Figure 3 genes-14-01756-f003:**
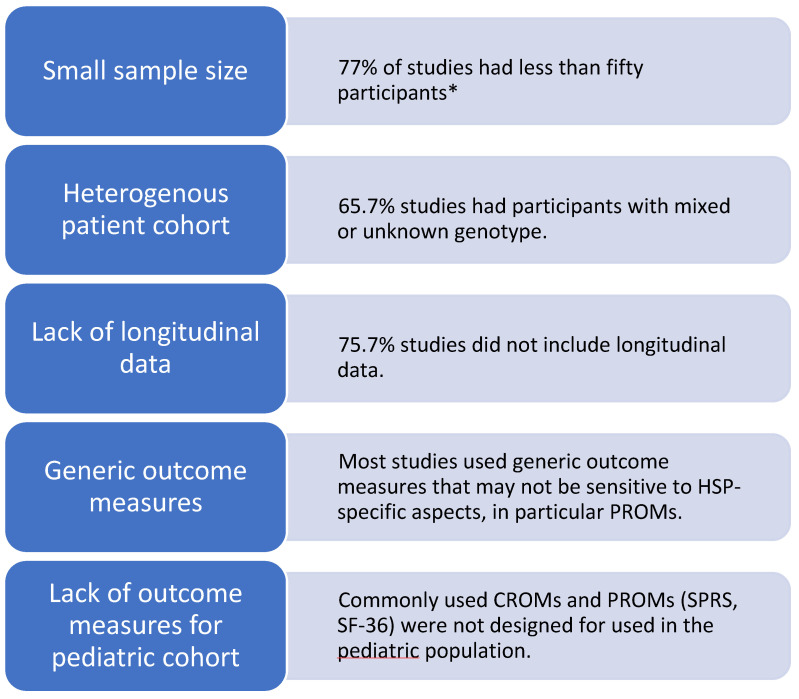
Common limitations of studies of HSP outcome measures. * [66] A sample size of less than fifty participants is defined as small in the COSMIN checklist. However, a smaller sample size may be justified in the study design. For example, a randomized controlled trial of 14 patients with *HSP-CYP7B1* calculated an estimate of the effect size of treatment using a pre-specified biomarker to determine the sample size required for adequate power [35].

**Table 1 genes-14-01756-t001:** Descriptive findings.

Study Type	Number of Studies (%)
Observational (total) Case series Cohort Case–control Cross-sectional	55 (78.6) 9 14 2 30
Diagnostic test accuracy	1 (1.4)
Non-randomized interventional studies	7 (10)
Randomized controlled trials	6 (8.6)
**Study Characteristics**	**N (%)**
Longitudinal data presented	17 (24.3)
Control group included	34 (48.6)
Intervention	18 (25.7)
**Participant Genotype**	**N (%)**
Single genotype	24 (34.3)
Mixed known and unknown genotypes	35 (50)
Unknown genotype	11 (15.7)
**Participant Characteristics**	**N (%)**
Number of participants	37.0 people (range 2–239, SD 42.2)
Mean of mean ages (n = 64)	39.9 years (range 4.8–62, SD 14.9)

**Table 2 genes-14-01756-t002:** Clinician-reported outcome measures.

Outcome Measure	Studies ^1^	Control Group (Patient vs. Control Difference) ^2^	Longitudinal (Disease Progression) ^3^	Intervention (Response to Intervention) ^4^
HSP-specific CROM				
SPRS	27	12 (2)	8 (6)	5 (1)
SPATAX-EUROSPA disability score	4	1 (0)	1 (1)	1 (0)
CROM for other neurological disorders				
SARA	3	1 (1)	1 (1)	0 (0)
ALSFRS-R	2	1 (0)	2 (1)	0 (0)
Unified Huntington’s Disease Rating Scale Part IV	1	1 (0)	0 (0)	0 (0)
Multiple Sclerosis Impairment Scale	1	0 (0)	0 (0)	1 (0)
Generic CROM				
Functional questionnaire score	1	0 (0)	0 (0)	1 (0)
Modified Rankin Scale	1	1 (0)	1 (0)	0 (0)
Disability Score (DIS)	1	0 (0)	0 (0)	1 (0)
Functional Independence Measure (FIM)	1	0 (0)	0 (0)	0 (0)

^1^ Total number of studies; ^2^ Number of studies that included a control group (Studies where outcome measure showed a difference between patients and controls); ^3^ Number of studies that had longitudinal data (studies where outcome measure was able to show disease progression); ^4^ Number of studies that included an intervention (Studies where outcome measure showed a response to the intervention). Details of the included studies are in Appendix A.

**Table 3 genes-14-01756-t003:** Performance outcome measures.

Outcome Measure	Studies ^1^	Control Group (Patient vs. Control Difference) ^2^	Longitudinal (Disease Progression) ^3^	Intervention (Response to Intervention) ^4^
Motor Function PerfOMS				
Modified Ashworth Scale	18	2 (0)	3 (1)	13 (11)
10MWT, 6MWT, 2MWT, 5MWT, 20MWT, 3-min endurance walk	14	3 (2)	1 (1)	9 (3)
Timed Up-and-Go test (TUG)	6	3 (2)	2 (2)	3 (1)
Medical Research Council muscle strength	4	0 (0)	0 (0)	2 (1)
Gross motor function measure (GMFM-66, GMFM-88)	2	0 (0)	0 (0)	2 (1)
Gross motor function classification score (GMFCS)	2	1 (0)	0 (0)	1 (0)
Falls Efficacy Scale-International (FES-I)	2	2 (1)	1 (0)	0 (0)
Berg balance scale	2	0 (0)	0 (0)	2 (1)
Composite cerebellar functional severity score (CCFSw)	2	1 (1)	1 (0)	0 (0)
Nine-hole pegboard test, click test, writing test, tapping test	2	1 (0)	0 (0)	1 (1)
Physiological Cost Index	2	1 (0)	1 (0)	2 (0)
Four-stage functional scale of motor disability (4SMD or 4FMS)	2	0 (0)	0 (0)	0 (0)
Activities specific Balance Confidence scale (ABC)	1	0 (0)	0 (0)	1 (0)
Modified version of Gillette Functional Assessment Questionnaire	1	0 (0)	0 (0)	1 (1)
Walking Handicap scale	1	0 (0)	0 (0)	1 (1)
Lower extremity subclass of Fugl–Meyer assessment	1	0 (0)	0 (0)	1 (0)
Ambulatory score (AMB)	1	0 (0)	0 (0)	1 (0)
AMBUS	1	0 (0)	1 (0)	0 (0)
Spasm Frequency Scale	1	0 (0)	1 (1)	1 (1)
Strength with microFET 2 hand-held dynamometer	1	0 (0)	0 (0)	1 (1)
Walking ability (landmarks of disability)	1	0 (0)	0 (0)	0 (0)
Cognitive Function PerfOMs				
Montreal Cognitive Assessment	4	4 (0)	1 (0)	0 (0)
Wechsler Adult Intelligence Scale—revised	3	1 (0)	1 (1)	0 (0)
Mini-Mental State Exam	2	2 (0)	0 (0)	0 (0)
Addenbrooke’s Cognitive Exam	1	1 (0)	1 (0)	0 (0)
CANTAB assessment	1	1 (0)	0 (0)	0 (0)
VLMT, FWIT, TMT A/B, FAB, d2-R, RWT *	1	0 (0)	1 (1)	0 (0)

^1^ Total number of studies; ^2^ Number of studies that included a control group (Studies where outcome measure showed a difference between patients and controls); ^3^ Studies that had longitudinal data (studies where outcome measure was able to show disease progression); ^4^ Studies that included an intervention (Studies where outcome measure showed a response to the intervention); * VLMT—Verbal Learning and Memory Test, FWIT—Farbe-Wort-Interferenz Test, TMT A/B—Trail Making Test Part A and B, FAB—Frontal Assessment Battery, d2-R—revised d2 Test of attention, RWT—Regensburg Word Fluency Test. Details of the included studies are in Appendix A.

**Table 4 genes-14-01756-t004:** Patient-Reported Outcome Measures.

Outcome Measure	Studies ^1^	Control Group (Patient vs. Control Difference) ^2^	Longitudinal (Disease Progression) ^3^	Intervention (Response to Intervention) ^4^
Quality of Life PROMs				
SF-36, SF-12, RAND 36-Item Health Survey	7	3 (2)	1 (0)	2 (1)
EQ-5D	3	3 (1)	0 (0)	0 (0)
Becks Depression Inventory (BDI-V)	2	2 (1)	0 (0)	0 (0)
Visual analogue score	2	0 (0)	0 (0)	2 (1)
Patient Health Questionnaire (PHQ-9)	1	1 (0)	0 (0)	0 (0)
Modified Goal Attainment Scale (mGAS)	1	0 (0)	0 (0)	1 (1)
International Consultation of Incontinence Questionnaire (ICIQ)—LUTSqol	1	0 (0)	0 (0)	0 (0)
Cerebral Palsy QoL questionnaire (CPQoL)	1	0 (0)	0 (0)	1 (1)
Caregiver Priorities and Child Health Index of Life with Disabilities (CPCHILD)	1	0 (0)	0 (0)	0 (0)
Zung depression score	1	1 (1)	0 (0)	0 (0)
ICIQ-Short Form	1	0 (0)	0 (0)	0 (0)
Hospital Anxiety and Depression Scale	1	0 (0)	0 (0)	1 (1)
Other PROMs				
Brief pain inventory	3	2 (1)	0 (0)	1 (0)
Modified Fatigue Impact Scale (MFI)	2	1 (1)	0 (0)	1 (0)
Multidimensional fatigue inventory	1	1 (0)	0 (0)	0 (0)
Scale for Outcomes in Parkinson’s Disease for Autonomic Symptoms (SCOPA-AUT)	1	0 (0)	0 (0)	0 (0)
Numeric rating scale for pain	1	0 (0)	0 (0)	1 (1)

^1^ Total number of studies; ^2^ Number of studies that included a control group (Studies where outcome measure showed a difference between patients and controls); ^3^ Number of studies that had longitudinal data (Studies where outcome measure was able to show disease progression); ^4^ studies that included an intervention (Studies where outcome measure showed a response to the intervention). Details of the included studies are in Appendix A.

**Table 5 genes-14-01756-t005:** Biochemical Biomarkers.

Biomarker	Studies ^1^	Control Group (Patient vs. Control Difference) ^2^	Longitudinal (Disease Progression) ^3^	Intervention (Response to Intervention) ^4^
Serum and CSF NfL	6	6 (5)	1 (0)	0 (0)
Serum and CSF 25-OHC and 27-OHC	4	3 (2)	2 (0)	2 (2)
Blood(plasma) and CSF amino acids	2	1 (1)	0 (0)	1 (0)
Lipidomics: fibroblast and plasma	1	1 (1)	0 (0)	0 (0)
Citrulline	1	0 (0)	0 (0)	0 (0)
Glycosylceramide profile	1	0 (0)	0 (0)	0 (0)
Neurofilament heavy chain: CSF and serum	2	1 (1)	1 (0)	0 (0)
Autophagy-related protein (ATG9A) ratio	1	1 (1)	0 (0)	0 (0)
CSF Aβ 1–42, total tau, phospho tau	1	1 (0)	0 (0)	0 (0)
Mitochondrial DNA levels; Muscle biopsy	1	1 (1)	0 (0)	0 (0)
Cell morphomics	1	1 (1)	0 (0)	1 (1)
Scanning electron microscopy of hair shafts	1	1 (1)	0 (0)	0 (0)

^1^ Total number of studies; ^2^ Number of studies that included a control group (Studies where outcome measure showed a difference between patients and controls); ^3^ Studies that had longitudinal data (Studies where outcome measure was able to show disease progression); ^4^ Studies that included an intervention (Studies where outcome measure showed a response to the intervention). Details of the included studies are in Appendix A.

**Table 6 genes-14-01756-t006:** Neuroimaging Biomarkers.

Biomarker	Studies ^1^	Control Group (Patient vs. Control Difference) ^2^	Longitudinal (Disease Progression) ^3^	Intervention (Response to Intervention) ^4^
MRI brain and spine	17	8 (5)	4 (1)	1 (0)
DTI	7	7 (7)	2 (1)	0 (0)
MRS	4	3 (2)	1 (0)	0 (0)
Ioflupane Single Photon Emission Computed Tomography (SPECT)	1	0 (0)	0 (0)	0 (0)

^1^ Total number of studies; ^2^ Number of studies that included a control group (Studies where outcome measure showed a difference between patients and controls); ^3^ Studies that had longitudinal data (studies where outcome measure was able to show disease progression); ^4^ Studies that included an intervention (Studies where outcome measure showed a response to the intervention). Details of the included studies are in Appendix A.

**Table 7 genes-14-01756-t007:** Neurophysiology Biomarkers.

Biomarker	Studies ^1^	Control Group (Patient vs. Control Difference) ^2^	Longitudinal (Disease Progression) ^3^	Intervention (Response to Intervention) ^4^
MEPs	10	2 (1)	2 (0)	3 (0)
NCS/EMG	9	2 (0)	2 (0)	1 (0)
SSEP	5	1 (1)	0 (0)	0 (0)
VEP	2	0 (0)	0 (0)	0 (0)
BAEP	1	0 (0)	0 (0)	0 (0)

^1^ Total number of studies; ^2^ Number of studies that included a control group (Studies where outcome measure showed a difference between patients and controls); ^3^ Studies that had longitudinal data (Studies where outcome measure was able to show disease progression); ^4^ Studies that included an intervention (Studies where outcome measure showed a response to the intervention). Details of the included studies are in Appendix A.

**Table 8 genes-14-01756-t008:** Other Biomarkers.

Biomarker	Studies ^1^	Control Group (Patient vs. Control Difference) ^2^	Longitudinal (Disease Progression) ^3^	Intervention (Response to Intervention) ^4^
Laboratory and mobile gait analysis	8	4 (4)	1 (1)	4 (2)
Spectral-domain optical coherence tomography (SD-OCT)	4	0 (0)	2 (0)	0 (0)
Video-oculography and rotational chair testing	1	1 (1)	0 (0)	0 (0)
Goniometer	2	0 (0)	0 (0)	1 (1)
Instrumented dynamic balance assessment	1	0 (0)	0 (0)	1 (1)
Urodynamic assessment	1	0 (0)	0 (0)	0 (0)
Video supported posturography	1	0 (0)	1 (0)	0 (0)
Protocol for Evaluation of Acquired Speech Disorders (PADAF)	1	1 (1)	0 (0)	0 (0)

^1^ Total number of studies; ^2^ Number of studies that included a control group (Studies where outcome measure showed a difference between patients and controls); ^3^ Studies that had longitudinal data (Studies where outcome measure was able to show disease progression); ^4^ Studies that included an intervention (Studies where outcome measure showed a response to the intervention). Details of the included studies are in Appendix A.

**Table 9 genes-14-01756-t009:** Genotype-specific biomarkers.

Biomarker	Studies	Genotype	Finding
Serum hydroxycholesterols [35,43,44,45]	4	HSP-*CYP7B1*	Higher 25- and 27-OHC levels in patients compared to controls
Plasma citrulline [55,56]	2	HSP-*ALDH18A1*	Low citrulline levels in patients (n = 3 and n = 4)
Lipidomics [57]	1	HSP-*PCYT2*	Accumulation of plasma phosphatidylcholine [O] etherphospholipids in patients (n = 3) compared to controls (n = 20)
Glycosylceramide profile [58]	1	HSP-*GBA2*	Elevated glycosylceramide levels in affected patient and carrier parent
ATG9A ratio (automated high-throughput imaging) [59]	1	HSP-*AP-4*	Increase in ATG9A ratio (intracellular distribution of ATG9A in trans-Golgi network compared to the remainder of the cell) in patient fibroblasts (n = 18) compared to asymptomatic carriers (n = 14)
mtDNA levels [60]	1	HSP-*SPG7*	Reduced mtDNA levels from whole blood in patients (n = 27) and carriers (n = 5) compared to controls (n = 17).
Scanning electron microscopy of hair shafts [61]	1	HSP-*FAHN*	Subtle to pronounced longitudinal grooves in hair shafts from 4/4 patients and adhesive plaques in 3/4 patients compared to controls.

**Table 10 genes-14-01756-t010:** Advantages and disadvantages of commonly used outcome measures for hereditary spastic paraplegia.

Outcome Measure	Advantages	Disadvantages
SPRS	HSP-specific. Validated against other measures of disability. Cross-cultural validation [67,68]. Longitudinal measurement [30,33,35,69].	Not validated in a pediatric cohort. Inter-rater reliability only measured between two raters from same center [67]. Requires 10 m space and stairs to measure walking time.
SPATAX-EUROSPA disability score	HSP-specific.	Not validated. Does not measure pain, bladder or bowel function.
Modified Ashworth scale	Accessible. Moderate reliability [70].	Potential for inter-rater variation particularly for lower limb assessment [70]. Variation depends on when test is performed (e.g., the timing of anti-spasticity medication).
6MWT, 10MWT, TUG and variations	Accessible.	Not suitable for patients who are unable to mobilize. Potential variation depending on hallway lengths used due to time taken to change directions [71].
SF-36	Validated [72]. Published population norms [73]. Some versions easily accessible. Cross-cultural validity [73,74]. Demonstrated worse QoL in patients with HSP compared to controls [74,75].	Generic and does not address HSP-specific aspects of QoL.
EQ-5D	Validated [76]. Easily accessible. Cross-cultural validity [76]. Published population norms.	Generic and does not address HSP-specific aspects of QoL. Not as widely used as SF-36 in HSP population.
Serum and CSF neurofilament light chain (NfL)	Able to distinguish patients vs. controls, pre-symptomatic and symptomatic patients [29,77,78].	Non-specific, elevated in other conditions such as ALS, Alzheimer’s dementia, Parkinson’s Disease [79]. CSF collection requires an invasive procedure with potential adverse effects, such as a low-pressure headache.
27 and 25 hydroxycholesterol (OHC)	Specific to SPG5 (HSP-*CYP7B1*). Decrease in serum and plasma 27-OHC levels in response to atorvastatin [35,44]. Good diagnostic biomarker.	Biochemical response to treatment not reflected in clinical benefit as measured with SPRS. Role as prognostic or monitoring biomarker yet to be determined.
MRI brain and spine	Accessible. Volumetric analysis showed atrophy in certain parts of the brain or spine in patients with HSP [29,31,43,80].	Heterogenous findings between and within genotypes [81].
DTI	Abnormal in patients vs. controls (see results section). Correlate with other outcome measures [82,83,84]. Imaging can be performed with most MRI machines. Able to identify axonal damage not seen on MRI.	No longitudinal data. Requires analysis by experienced personnel.
MEPs	Able to measure upper motor neuron abnormalities seen in HSP [85]. Lower limb CMCT abnormal in 78% patients with HSP [86].	Inconsistent findings across various studies [86]. Requires specialized equipment and technical expertise. Some patients may find MEP studies uncomfortable.
NCS/EMG	Majority of patients with HSP-*SPG11* have axonal neuropathy [31,33].	Inconsistent findings across various studies.
Gait analysis	Able to differentiate patients vs. controls (see results). Able to demonstrate response to intervention [49,50]. Correlate with SPRS scores [28,34]. Mobile gait analysis system able to show disease progression over time [87].	Require expensive equipment. Data analysis can be complex depending on parameters used. Not suitable for participants who are unable to walk.
Retinal nerve fiber layer with OCT	Abnormal in 39% of patients with HSP [88].	No change in RNFL over time. No clinical correlation with SPRS. Inconsistent findings [88].

## Data Availability

All data supporting reported materials are available in the manuscript and Appendix A.

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
