# Peer review of "Outcome Measures and Biomarkers for Clinical Trials in Hereditary Spastic Paraplegia: A Scoping Review"

_genes, 2023, doi:10.3390/genes14091756_

Round 1

Reviewer 1 Report

In their review article „Outcome measures and biomarkers for clinical trials in hereditary spastic paraplegia: A scoping review“, Siow et al. give structured and carefully researched overview of existing data on outcome measures in HSP. The data are sound and clearly presented. The topic is of high relevance since HSP interventional trials are coming up.

One major issue should be considered: Several studies have been overlooked in the literature search, and the authors should consider including Amprosi et al. (PMID 36636734), Loris et al. (PMID 36622133) and Alecu et al. (PMID 37482941).

Minors:

„Although 12/27 studies included control groups, comparison of SPRS scores in patients versus controls was performed in only two studies, both showing significant differences (28, 29).“ It should be noted that measuring the SPRS in „healthy controls“ is only reasonable for comparison to pre-manifest or prodromal HSP, but not for clinically established HSP.

Supplemental table 3 should be improved by adding the exact numbers of patients and controls as well as the period of intervention.

The design of Figure 3 is misleading (why are the arrows pointing to the right?), it should either be changed into an infobox or a more clear figure.

Author Response

The authors would like to thank the reviewer for their time and effort in reviewing the manuscript. We thank the reviewer for their helpful comments. In response to the comments, we would like to respond with the below:

One major issue should be considered: Several studies have been overlooked in the literature search, and the authors should consider including Amprosi et al. (PMID 36636734), Loris et al. (PMID 36622133) and Alecu et al. (PMID 37482941).

These important studies regarding HSP biomarkers were published after the scoping review search was performed (see Appendix A) and therefore were not included in the search results. We have incorporated discussion of these recently published studies in the discussion section as follows:

Line 423-427 “We recommend the SPRS as it was the most commonly used CROM in HSP-related studies, has undergone psychometric testing in a cohort of individuals with HSP, has demonstrated cross-cultural validity, has been tested for responsiveness in longitudinal (Amprosi et al) and interventional studies, and is a disease-specific outcome measure.”

Line 439-441 “These findings are echoed by a recent study of CROMs and PROMs in HSP identifying the SPRS as a suitable CROM to measure disease progression and the need for an HSP-specific PROM (Amprosi et al).”

Table 10

Serum and CSF neurofilament light chain (NfL)

Able to distinguish patients vs controls, pre-symptomatic and symptomatic patients (Alecu et al)

Gait analysis

Mobile gait analysis system able to show disease progression over time (Loris et al)

Minors:

„Although 12/27 studies included control groups, comparison of SPRS scores in patients versus controls was performed in only two studies, both showing significant differences (28, 29).“ It should be noted that measuring the SPRS in „healthy controls“ is only reasonable for comparison to pre-manifest or prodromal HSP, but not for clinically established HSP.

We have addressed this with the following statement:

Line 170-172 “It is important to note that SPRS values from healthy controls are most relevant when comparing to pre-symptomatic HSP carriers rather than for comparison to individuals with symptomatic HSP.”

Supplemental table 3 should be improved by adding the exact numbers of patients and controls as well as the period of intervention.

The number of patients is included in the row “number of participants”, there were no controls as six out of eight studies were crossover studies, whilst the other two studies had equal distribution of participants in intervention and placebo/sham group as detailed in the “Intervention” row.

We have included an additional row for “Duration of intervention”.

The design of Figure 3 is misleading (why are the arrows pointing to the right?), it should either be changed into an infobox or a more clear figure.

The design of Figure 3 has been changed to remove the arrows.

Reviewer 2 Report

A scoping review by Siow et al is devoted to the analysis of known criteria for evaluation and biomarkers  for clinical trials in hereditary spastic paraplegia. The review is written in compliance with the rules required for this kind of scientific publications, with a detailed description of the search algorithms, goals and selection criteria. Based on the presented data, the authors propose recommendations for HSP clinical trial design.

I would like to make a few comments on the text of the work.

The main remark is whether the presented work corresponds to the subject of the journal "Genes" and in particular Special Issue: Study on Genotypes and Phenotypes of Neurodegenerative Diseases. The authors do not consider different evaluation criteria and biomarkers depending on the genotype, although such studies are available. When conducting an HSP clinical trial, it is necessary to take into account the genotype of patients whenever possible. Therefore, it is necessary to add the appropriate section/sections to the manuscript.

-         Figure 1. “Record excluded**”. What does ** mean? Should be explained or deleted.

-         Tables 2 - 6 should indicate in which studies these parameters are presented.

-         Figure 3. The reference to (59) (“Small sample size” limitation) is not entirely clear. Indeed, in (59) it is stated that “The suggested sample size requirements should be considered as the basic rules; in some situations, dependent on the type of model, number of factors or items, more nuanced criteria might be applied. For example, a smaller sample size might be acceptable when an argument is presented in the individual study, stating the considerations why a smaller sample size is adequate. Subsequently, the study can still be rated as very good or adequate, despite lower sizes than requested in the standard”.

Minor editing of English language

Author Response

The authors would like to thank the reviewer for their time and effort in reviewing the manuscript. In response to the comments, we would like to respond with the below:

The main remark is whether the presented work corresponds to the subject of the journal "Genes" and in particular Special Issue: Study on Genotypes and Phenotypes of Neurodegenerative Diseases. The authors do not consider different evaluation criteria and biomarkers depending on the genotype, although such studies are available. When conducting an HSP clinical trial, it is necessary to take into account the genotype of patients whenever possible. Therefore, it is necessary to add the appropriate section/sections to the manuscript.

We thank the reviewer for this helpful suggestion. We have added the section “3.4.5 Genotype-specific biomarkers in HSP” and Table 9 to the results. We suggest that the article is now a better fit for the Special Issue topic.

-         Figure 1. “Record excluded**”. What does ** mean? Should be explained or deleted.

“**” has been deleted from Figure 1.

-         Tables 2 - 6 should indicate in which studies these parameters are presented.

Information regarding which studies theses parameters are presented in are included in Supplementary Material 2 to manage the volume of information in the manuscript. We have now included more information in the footers for Tables 2-8 “Details of studies included are in Supplementary Material 2.”.

-         Figure 3. The reference to (59) (“Small sample size” limitation) is not entirely clear. Indeed, in (59) it is stated that “The suggested sample size requirements should be considered as the basic rules; in some situations, dependent on the type of model, number of factors or items, more nuanced criteria might be applied. For example, a smaller sample size might be acceptable when an argument is presented in the individual study, stating the considerations why a smaller sample size is adequate. Subsequently, the study can still be rated as very good or adequate, despite lower sizes than requested in the standard”.

The authors have clarified this in the footnote.

*(59) A sample size of less than fifty participants is defined as small in the COSMIN checklist. However, a smaller sample size may be justified in the study design. For example, a randomized controlled trial of 14 patients with HSP-CYP7B1 calculated an estimate of the effect size of treatment using a pre-specified biomarker to determine the sample size required for adequate power (35).

Reviewer 3 Report

The authors have presented an interesting paper about potential biomarkers and outcome measures in HSP.

This report is well written and extensive. 

The results are significant and novel.

All references are up-to-date and well cited.

No further comments and suggestions.

Author Response

The authors would like to thank the reviewer for their time and effort in reviewing the manuscript. We thank the reviewer for their positive comments.

Round 2

Reviewer 2 Report

Agree. Thank you.

Minor editing of English language